# Hierarchically Porous Silk/Activated-Carbon Composite Fibres for Adsorption and Repellence of Volatile Organic Compounds

**DOI:** 10.3390/molecules25051207

**Published:** 2020-03-07

**Authors:** Aled D. Roberts, Jet-Sing M. Lee, Adrián Magaz, Martin W. Smith, Michael Dennis, Nigel S. Scrutton, Jonny J. Blaker

**Affiliations:** 1Bio-Active Materials Group, Department of Materials & Henry Royce Institute, The University of Manchester, Manchester M13 9PL, UK; aled.roberts@manchester.ac.uk (A.D.R.); adrian.magaz@postgrad.manchester.ac.uk (A.M.); 2Future Biomanufacturing Research Hub (FBRH), Manchester Institute of Biotechnology, The University of Manchester, Manchester M1 7DN, UK; nigel.scrutton@manchester.ac.uk; 3Institute for Integrated Cell-Material Sciences, Institute for Advanced Study, Kyoto University, Yoshida, Sakyo-ku, Kyoto 606-8501, Japan; lee@macro.t.u-tokyo.ac.jp; 4Defence Science and Technology Laboratory (Dstl), Porton Down, Salisbury SP4 0JQ, UK; MWSMITH@dstl.gov.uk (M.W.S.); MRDENNIS@mail.dstl.gov.uk (M.D.)

**Keywords:** porous fibres, activated carbon, ice-templating, ice segregation induced self-assembly, silk fibroin, wet spinning, solution blow spinning, freeze casting

## Abstract

Fabrics comprised of porous fibres could provide effective passive protection against chemical and biological (CB) threats whilst maintaining high air permeability (breathability). Here, we fabricate hierarchically porous fibres consisting of regenerated silk fibroin (RSF) and activated-carbon (AC) prepared through two fibre spinning techniques in combination with ice-templating—namely cryogenic solution blow spinning (Cryo-SBS) and cryogenic wet-spinning (Cryo-WS). The Cryo-WS RSF fibres had exceptionally small macropores (as low as 0.1 µm) and high specific surface areas (SSAs) of up to 79 m^2^·g^−1^. The incorporation of AC could further increase the SSA to 210 m^2^·g^−1^ (25 wt.% loading) whilst also increasing adsorption capacity for volatile organic compounds (VOCs).

## 1. Introduction

The current generation of air-permeable chemical and biological (CB) resistant clothing, such as Joint Service Lightweight Integrated Suit Technology (JSLIST), consists of a particulate activated carbon (AC) adsorbent layer laminated between multiple layers of fabric—including a hydrophobic outer layer and soft inner layer [1,2]. Combining the properties of these layers into a single material could reduce the physiological burden on the user through mass reduction and enhanced diffusive and evaporative heat transport. Fabrics comprised of porous fibres with a significant capacity to adsorb and repel volatile organic compounds (VOCs) could help achieve this goal. In particular, a hierarchical structure of pores—where networks of larger macropores (> 50 nm) provide effective diffusion and access to smaller meso- (2–50 nm) and micropores (< 2 nm)—could provide effective rapid adsorption activity whilst maintaining high permeability and relatively low mass [3]. Furthermore, fibres with micron-scale surface features could also enhance liquid repellence through a ‘lotus-leaf’ type effect—repelling liquid VOC droplets as well as dirt, dust and mud [4].

Although other highly porous materials such as metal organic frameworks (MOFs) and silicates have demonstrated exceptional adsorption and degradation activity against various chemical warfare agents (CWAs), [1,5,6,7] their relatively poor chemical and physical stability and high cost of manufacture mean they are unlikely to replace low-cost ACs (which can also effectively catalyse the degradation of CWAs [8]) for general applications. The brittle nature and poor mechanical properties of microporous materials such as ACs, MOFs and silicates make them unsuitable for use as stand-alone materials in fabrics; instead, they need to be combined with another material with suitable physical properties in order to be incorporated into a garment. This can be achieved by decorating the surface of fibres with a porous material, [1,5,6] but loss of physical attachment means durability can be a significant issue.

In this work, we reprocess silkworm silk into aqueous regenerated silk fibroin (RSF) solutions before spinning into porous fibres via two cryogenic spinning techniques, namely Cryogenic Solution Blow Spinning (Cryo-SBS) and Cryogenic Wet Spinning (Cryo-WS). Cryogenically freezing hydrated fibres results in ice-segregation-induced self-assembly (ISISA) which, after freeze-drying, produces a macroporous ice-templated structure (Figure 1) [9,10,11,12,13,14,15,16,17,18]. By spinning colloidal suspensions of AC in RSF, macroporous fibres loaded with AC can be obtained—increasing the specific surface area (SSA), hydrophobicity and adsorption capacity for VOCs. A summary of the conditions employed to prepare samples, along with the most relevant physical properties, is given in Table 1.

## 2. Results

RSF spinning dopes were prepared following a previously published protocol detailed in the SI [11,19]. Porous RSF fibres were prepared via Cryo-SBS (Figure 1a) following a protocol previously developed within our group, [10,11,20] however, the Cryo-WS method (Figure 1b) has, to the authors’ knowledge, not been reported previously. For the Cryo-SBS method, RSF and colloidal RSF-AC mixtures were spun directly into a liquid nitrogen (LN_2_) bath, collected and freeze dried (Figure 1a). This rapid freezing induces the formation of macropores via ISISA [9,10,11,12,13,14]. The obtained fibres were approximately 0.2–1 cm in length and 25–60 µm in diameter, and darkened uniformly with increasing AC content (Figure 1c). Fibres with an AC content greater than 10 wt.% could not be produced due to instability of the fibre jet—resulting in droplet formation rather than a continuous stream of fibres. Their relatively short length meant they would be unsuitable for processing into a woven fabric, but could potentially be employed as a non-woven material (i.e., a felt). Scanning electron microscopy (SEM) imaging revealed macropores with average diameters ranging from 1–13 µm in diameter (Figure 1c), which are fairly small for a freeze-casting method—which typically produces pores in the range of 5–50 µm [21,22] The relatively small pores are likely due to rapid freezing from the high surface area-to-volume ratio of the fibres and high temperature differential between the fibres and LN_2_, since the pore sizes from freeze-casting methods are inversely proportional to the velocity of ice crystal growth [22]. It can also be observed from the SEM images that the fibres have a seemingly non-porous outer sheath. This likely arises from evaporation-induced phase separation at the surface of the fibre as it travels through the air gap between the nozzle and the cryogenic bath; a phenomenon also observed for some dry-jet wet spun fibres [23]. For the Cryo-WS method, RSF and colloidal RSF-AC mixtures were wet-spun into an ethanol coagulation bath akin to previous literature reports for wet-spun RSF fibres (Figure 1b) [24,25,26,27,28]. The wet fibres were then subjected to solvent exchange in deionised (DI) water, before being rapidly frozen by submersion in LN_2_ followed by freeze-drying. The dry fibres were continuous in length and had diameters ranging from 100–220 µm (Figure 1d); smaller diameter fibres could feasibly be produced by employing a smaller diameter extrusion nozzle or by employing post-spin drawing. The fibres were fairly flexible to handle and also darkened with increasing AC content. AC loadings in excess of 25 wt.% could not be achieved since these fibres’ mechanical properties after spinning were too poor and could not be collected without breaking apart. The fibres had average macropore diameters ranging from 0.1–1 µm, which—to the author’s knowledge—are among the smallest reported for an ice-templating based method (where water is the solvent). This is likely due to the rapid freezing resulting in relatively small ice crystals and hence small pores after freeze-dying [22]. Xia and co-workers produced porous fibres with similarly small (~ 0.1 µm) macropores via a cryogenic electrospinning technique (employing organic solvents rather than water), suggesting a similar underlying pore formation mechanism [13].

The SSA and micropore volume of the fibres were determined through N_2_ gas sorption and Brunauer–Emmett–Teller (BET) analysis (Figure 2 and Table 1). The surface area of the Cryo-SBS fibres with 0% AC was 34 m^2^·g^−1^, which is fairly high for ISISA-derived pores which typically have SSAs less than 15 m^2^·g^−1^ [21,22]. This relatively high surface area is likely a result of the relatively small macropores as observed by SEM. A 4% AC loading saw a decrease in SSA to 17 m^2^·g^−1^, only 28% of the theoretical SSA considering the masses of the silk (at 34 m^2^·g^−1^) and AC (at 697 m^2^·g^−1^) individually, suggesting that most pores were inaccessible. A 10% AC loading yielded an SSA of 44 m^2^·g^−1^, which was 44% of the theoretical SSA (100 m^2^·g^−1^). This relatively poor performance could be attributed to the outer non-porous sheath restricting gaseous diffusion to the AC, or rapid freezing resulting in smothering of the AC particles within the RSF polymer. Analysis of the isotherms revealed an increase in micropore volume from 0.038 to 0.05 cm^3^·g^−1^ with a 10% AC loading, suggesting the increase in SSA was largely from the microporous AC. The Cryo-WS fibres with 0% AC had a significantly higher SSA at 79 m^2^·g^−1^ which—to the best of our knowledge—is the highest reported SSA for an ice-templated polymeric material. Incorporation of AC gradually increased the SSA to a maximum value of 210 m^2^·g^−1^ with a 25% AC loading—corresponding to an accessible surface area of 90% (Figure 2c). Use of ACs with higher SSAs (the employed AC had a SSA of 697 m^2^·g^−1^, but some ACs can exceed 4000 m^2^·g^−1^) [29] or other highly microporous materials such as MOFs would likely result in fibres with significantly higher SSAs. 

The efficacy of the fibres for adsorption of VOCs was assessed through Dynamic Vapour Sorption (DVS); here, cyclohexane was employed as a representative VOC for proof of principle. Porous fibres produced via Cryo-WS displayed significant adsorption capacity for VOCs (Table 1, Figure 2d.), with the majority of adsorption occurring at relatively high partial pressures—suggesting macropores from ISISA were largely contributing to adsorption rather than micropores from the AC. This was corroborated by the fact that an increase in AC loading from 10% to 25% had little effect on cyclohexane adsorption, although 0% AC loading had a smaller adsorption capacity. No significant cyclohexane adsorption could be detected for the Cryo-SBS fibres however, possibly due their lower SSAs or the presence of the non-porous sheath outer restricting diffusion to the porous core.

The mechanical properties for the fibres were investigated through uniaxial tensile testing (Appendix A), however the highly porous nature of the fibres meant the true cross-sectional surface area of the fibres (needed to convert force to stress) could not be accurately determined, and therefore calculation of the tensile strength was not possible. Contact angle measurements on RSF fibre mats produced by Cryo-SBS with 0% and 4% AC loading revealed increased hydrophobicity with an increasing AC content (Appendix A.). This is likely due to the relatively hydrophobic nature of AC on the surface of the fibres reducing their wettability, but may also be a result of enhanced surface roughness causing a “lotus-leaf” type repellence effect (Appendix A)—a more in depth study is needed to confirm this possibility however. Contact angle data could not be obtained for Cryo-WS fibres since their relatively bulky nature meant adequately uniform mats could not be produced. Visible light and cross-polarised microscopy was also performed on the fibres to visualise the distribution of AC, a technique which was recently employed to visualise the distribution of graphene within hollow aramid fibres (Appendix A) [30]. This revealed a fairly uniform distribution of AC within the fibres (i.e., no significant agglomeration observed).

The composition and microstructure of the dry fibres were analysed through wide angle X-ray diffraction (WAXD) and Fourier-transform infrared spectroscopy (FTIR). Natural silk has a polycrystalline structure arising from ordered β-sheet domains, which has also been observed in some RSF-derived materials [24,25,26]. The obtained WAXD patterns were, however, indistinguishable from a background measurement, suggesting low crystallinity within these porous fibres (Appendix A). The high void volume in the porous fibres could also account for the low WAXD signal strength. FTIR was employed to probe the protein secondary structure of the RSF fibres through analysis of the amide I region of the spectrum (1600–1700 cm^−1^), allowing determination of the relative percentage of secondary structural features (i.e., random coils, α-helixes, β-sheets and β-turns). This revealed a high content of amorphous features relative to crystalline β-sheets, in concordance with the WAXD data. Post-treatment of the RSF fibres with ethanol, which has been shown to induce β-sheet formation, [11] was also performed—resulting in a significant increase (approx. 1.7-fold) in β-sheet features for the porous RSF fibres from both fibre spinning techniques. Post-treatment with ethanol could therefore be exploited to moderate the mechanical properties of the fibres, since a higher β-sheet content typically results in stronger, stiffer fibres [31].

## 3. Conclusions

In this work, hierarchically porous RSF fibres loaded with AC were produced through two cryogenic fibre spinning techniques—namely Cryo-SBS and a novel Cryo-WS method. Both of these techniques produced fibres with significantly different properties; notably Cryo-SBS fibres had a porous core and non-porous outer sheath which likely restricted gaseous diffusion, compromising accessible SSA and hence VOC adsorption capacity, these fibres were also discontinuous in length. Cryo-WS fibres, on the other hand, were continuous in length and porous throughout, with exceptionally small macropores for an ice-templating method (c.a., 0.1 µm) with significant VOC adsorption capacity. Further refinement of the spinning techniques may produce mesoporous (0.05 µm) fibres with further improved VOC adsorption, and the use of ultra-high surface area ACs or other microporous materials may also offer improvements. With further development, fibres such as these could be incorporated into woven or non-woven materials where their hierarchically porous structure in tandem with high VOC adsorption capacity could provide passive protection against CWAs or other airborne toxins. Drawbacks of such fibres include relatively poor mechanical properties, difficulty of dying/colouration and higher costs than materials typically employed in military uniforms (e.g., 50:50 cotton and polyester blends); but these issues could potentially be mitigated by bonding/knitting the porous fabrics as backing or lining layers to existing established materials. Porous RSF fibres such as these, loaded with other functional substances rather than AC, could have applications in a range of fields, particularly tissue regeneration and controlled drug delivery [32,33,34,35]. Finally, the use of genetically engineered silks such as recombinant spider silk could tune the mechanical and textural properties of the silk, or introduce chemical functionality such as additional lysine groups (-NH_2_) to neutralise toxic VOCs or other substances [36].

## Figures and Tables

**Figure 1 molecules-25-01207-f001:**
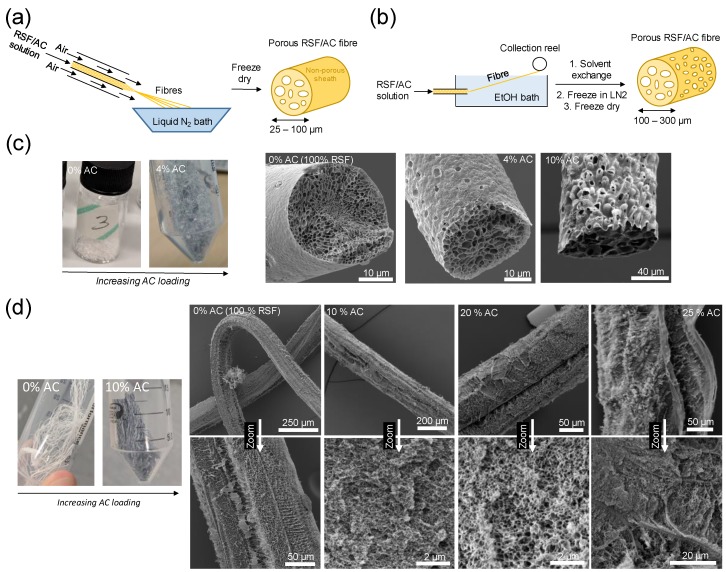
Schematic representation of (**a**) the cryogenic solution blow spinning (Cryo-SBS) and (**b**) the cryogenic wet-spinning (Cryo-WS) fibre spinning rigs. (**c**) Visible light and SEM images of RSF fibres produced by Cryo-SBS with increasing AC loading (left to right, 0, 4 and 10 wt. %). (**d**) Visible light and SEM images of RSF fibres produced by Cryo-WS with increasing AC loading (left to right, 0, 10, 20 and 25 wt. %).

**Figure 2 molecules-25-01207-f002:**
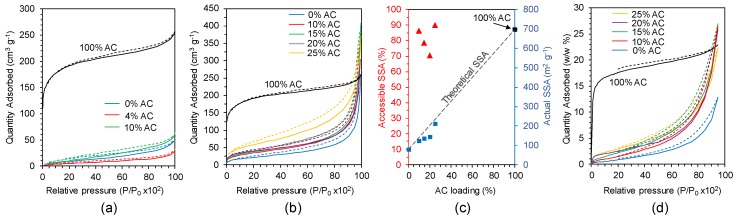
N_2_ gas sorption isotherms for porous RSF fibres produced by (**a**) Cryo-SBS with 0, 4 and 10 wt.% AC loading and (**b**) Cryo-WS with 0, 10, 15, 20 and 25 wt. % AC loading. Dashed lines indicate desorption, 100% AC included for comparison. (**c**) Relationship between AC loading and both % accessible SSA and actual SSA for RSF fibres produced by Cryo-WS. (**d**) Cyclohexane adsorption isotherms for RSF fibres produced through Cryo-WS with 0–25 wt. % AC loading. 100% AC included for comparison.

**Table 1 molecules-25-01207-t001:** Summary of the physical properties of the porous regenerated silk fibroin (RSF) and RSF/activated carbon (AC) fibres produced. Approximate fibre and macropore diameters (Ø) determined through multiple measurements from SEM images (± standard deviation).

Method	AC Loading [wt.%]	Average Fibre Ø [µm]	Average Macropore Ø [µm]	N_2_ BET SSA [m^2^·g^−1^]	Accessible SSA [%]	Micropore Vol. [cm^3^·g^−1^]	Max. Cyclohexane Uptake [*w*/*w*%]
(AC only)	100	n/a	n/a	697	100	0.358	22.8
Cryo-SBS	0	43 ± 9	1.1 ± 0.4	34	100	0.038	n/a
Cryo-SBS	4	23 ± 4	1.8 ± 0.5	17	28	0.015	n/a
Cryo-SBS	10	55 ± 6	10.4 ± 3.1	44	44	0.051	n/a
Cryo-WS	0	160 ± 25	1.6 ± 1.0	79	100	0.032	12.9
Cryo-WS	10	124 ± 14	0.1 ± 0.04	121	86	0.035	26.6
Cryo-WS	15	110 ± 16	0.68 ± 0.30	134	79	0.018	27.0
Cryo-WS	20	180 ± 30	0.14 ± 0.05	143	71	0.047	25.5
Cryo-WS	25	190 ± 29	0.54 ± 0.45	210	90	0.156	22.4

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
