# Peer review of "Hierarchically Porous Silk/Activated-Carbon Composite Fibres for Adsorption and Repellence of Volatile Organic Compounds"

_molecules, 2020, doi:10.3390/molecules25051207_

Round 1
Reviewer 1 Report
This article needs a major review before publication. The problems that should be solved are indicated below:
-Silk products are usually used in fashion due to the remarkable properties of clothing: low weight, softness, hydrophilicity and pleasing colors after bleaching / dyeing. Activation with C of the fibers worsens all these characteristics, transforming it into a hydrophobic fiber capable of adsorbing VOC. The natural question is where can the products obtained from these carbon activated silk fibers be used ? The uses indicated in the Conclusions section are neither correct nor realistic.
-The idea of functionalization of a fiber in order to increase the porosity is good, but it would be advisable to use functionalization agents that add value to the silk, but does not diminish it.
-How was carbon activation of the RSF carried out?
-What are the benefits of AC fiber? Silk is known to be soft and usually bleached or colored in vivid colors, following the processes of textile chemical finishing; how do you think these operations will be carried out, after the weaving process if the fibers become gray-black after the AC operation?
--Conclusion section: lines 170-177 should be reformulated. Generally, in the Conclusions section, the results obtained by the authors are highlighted, and not the results indicated by the literature. By functionalizing silk through AC, there is no protection against CWAs as long as the functionalized fibers have a higher VOC adsorption capacity and a low desorption capacity (Fig.2), thus a part of the toxic gases that remains in the fibers' pores in a fabric will be closer to the human body.
Author Response
Reviewer 1
Comments and Suggestions for Authors
This article needs a major review before publication. The problems that should be solved are indicated below:
-Silk products are usually used in fashion due to the remarkable properties of clothing: low weight, softness, hydrophilicity and pleasing colors after bleaching / dyeing. Activation with C of the fibers worsens all these characteristics, transforming it into a hydrophobic fiber capable of adsorbing VOC. The natural question is where can the products obtained from these carbon activated silk fibers be used ?
We agree with the reviewer that the silk/activated-carbon fibers would not be any good from a fashion perspective for the reasons they suggested, however we make no claim in the manuscript that they would be. As clearly detailed in the first paragraph of the introduction, the proposed application of these porous silk/activated-carbon fibers is for protection against toxic volatile organic compounds (VOCs), not for fashion/aesthetics purposes.
- The uses indicated in the Conclusions section are neither correct nor realistic.
The potential other uses detailed in the conclusion (i.e., tissue regeneration and drug delivery) are supported by a number of references (refs 32-35) which have employed other kinds of porous silk materials for such applications (for example, Guan, G. et al., Promoted dermis healing from full-thickness skin defect by porous silk fibroin scaffolds (PSFSs). Biomed. Mater. Eng. 2010, 20, 295–308.). Hence, we disagree with the reviewer and argue that the proposed possible applications are indeed realistic for future work.
-The idea of functionalization of a fiber in order to increase the porosity is good, but it would be advisable to use functionalization agents that add value to the silk, but does not diminish it.
We thank the reviewer for the helpful comment, and have modified the sentence on lines 174-175 to note that “other functional substances rather than AC” could be loaded into the porous fibers to enhance the properties for other applications.
-How was carbon activation of the RSF carried out?
Carbon activation was not done in-house. Instead, pre-activated carbon was purchased directly from a supplier as detailed in the Source Materials section of the SI.
-What are the benefits of AC fiber? Silk is known to be soft and usually bleached or colored in vivid colors, following the processes of textile chemical finishing; how do you think these operations will be carried out, after the weaving process if the fibers become gray-black after the AC operation?
As stated in the conclusions section (lines 176 – 178), the main benefit of the fibers would be (with further development) a fabric that could provide effective passive protection against airborne toxins. The reviewer is right in that the fibres could probably not be bleached or vividly coloured effectively with a high AC content due to its darkening effect, but in our view the colouration of the fabric is something to be considered significantly further down the development pathway and is beyond the scope of this communication – especially considering the intended application it not fashion/aesthetics.
--Conclusion section: lines 170-177 should be reformulated. Generally, in the Conclusions section, the results obtained by the authors are highlighted, and not the results indicated by the literature.
After reiterating our results in the conclusions section, we speculate about other possible applications and potential future work between the specified lines (now 179-184). We do not detail results in the literature as the reviewer suggests.
- By functionalizing silk through AC, there is no protection against CWAs as long as the functionalized fibers have a higher VOC adsorption capacity and a low desorption capacity (Fig.2), thus a part of the toxic gases that remains in the fibers' pores in a fabric will be closer to the human body.
We disagree with the reviewer’s interpretation. The fact that activated carbon is already used in widely-adopted JSLIST uniform (as stated in the first sentence of the introduction) to protect against toxic gasses shows that having AC close to the human body serves to protect against toxic gasses.
Reviewer 2 Report
Questions
- Authors have not clearly defined the materials and methods legend. Kindly explain it.
- Authors should emphasize the reason why Cryo-SBS has a AC loading max of 10% and Cryo-WS has a AC loading max of 25%?
- Authors should briefly provide the explanation regarding the phenomena of rupturing during the rapid freezing?
- Authors should provide more information regarding the pore size in the manuscript for the prepared fibers
Comments:
- Authors should precisely mention the instrument specifications.
- Overall revision of the spacing and the vocabulary is required.
- Abbreviations should be corrected.
Author Response
Reviewer 2
Comments and Suggestions for Authors
Questions
-Authors have not clearly defined the materials and methods legend. Kindly explain it.
The headings and legends in the SI have now been modified for clarity. We now specify an “Experimental details” section in addition to “Source Materials” and “Characterisation Techniques”. Legends in Figure S1 also changed slightly to improve clarity.
-Authors should emphasize the reason why Cryo-SBS has a AC loading max of 10% and Cryo-WS has a AC loading max of 25%?
This has now been done. For Cryo-SBS, the following sentence has been included from line 70: “Fibres with an AC content greater than 10 wt. % could not be produced due to instability of the fibre jet – resulting in droplet formation rather than a continuous stream of fibres.”
For Cryo-WS, the following sentence has been included from line 89: “AC loadings in excess of 25 wt. % could not be achieved since these fibres’ mechanical properties after spinning were too poor and could not be collected without breaking apart.”
-Authors should briefly provide the explanation regarding the phenomena of rupturing during the rapid freezing?
Not clear what the reviewer is referring to, but the phenomenon of pore formation arising from rapid freezing is described in the sentence beginning on line 53 – with references to numerous literature reports to provide greater detail if required by the reader.
-Authors should provide more information regarding the pore size in the manuscript for the prepared fibers
Table 1 has now modified to show the average macropore size and standard deviation, rather than the approximate pore size range.
Comments:
-Authors should precisely mention the instrument specifications.
We have already provided detailed specifications of the instruments used in the SI, we aren’t aware of any further details we can provide.
-Overall revision of the spacing and the vocabulary is required.
The vocabulary has been revised but only a minor typo was found (line 169), not clear what other spacing or vocabulary issues the reviewer is referring to.
-Abbreviations should be corrected.
Not clear which abbreviations the reviewer is referring to, we could not find any incorrect abbreviations.
Reviewer 3 Report
The paper submitted by Roberts and co-workers is about the fabrication of hierarchically porous silk/activated-carbon composite fibers for adsorption and repellence of volatile organic compounds. The obtained results are interesting and useful. Authors presented the experimental evidence in a clear and organized manner. However, there are some items that need to be addressed before this work is suitable for publication:
- In Table 1, authors should report the standard deviation of each value.
- In Table, the range of macropore diameter for each method is so wide. Authors should report histograms of macropore diameter for each method.
- Authors should include the meaning of the acronyms such as SSA, ISISA and others.
- Why did some composites present higher cyclohexane uptake than AC if AC has so much higher SSA?
Author Response
Reviewer 3
Comments and Suggestions for Authors
-The paper submitted by Roberts and co-workers is about the fabrication of hierarchically porous silk/activated-carbon composite fibers for adsorption and repellence of volatile organic compounds. The obtained results are interesting and useful. Authors presented the experimental evidence in a clear and organized manner. However, there are some items that need to be addressed before this work is suitable for publication:
-In Table 1, authors should report the standard deviation of each value.
This has been done where applicable. Data from N2/cyclohexane adsorption measurements could not be given a standard deviation due to only one measurement taken per sample. Further measurements could not be obtained to long data acquisition time (approx. 24 hours) and equipment access limitations.
-In Table, the range of macropore diameter for each method is so wide. Authors should report histograms of macropore diameter for each method.
Table 1 has now been changed to report the average macropore diameter with standard deviation, rather than the approximate size range.
-Authors should include the meaning of the acronyms such as SSA, ISISA and others.
Acronym for SSA given on line 56, acronym for ISISA given on line 54.
-Why did some composites present higher cyclohexane uptake than AC if AC has so much higher SSA?
This is due to the macropores of the silk having cyclohexane adsorption capacity, demonstrated by the fact that Cryo-WS silk with 0 % AC loading still has a significant cyclohexane uptake of 12.9 %. This explanation is given in the paragraph between lines 128 and 136.
Round 2
Reviewer 1 Report
From the first reading of the article I understood what purpose your modified fibers have; my observation was that it is unusual to use silk as a textile support for military uniforms. In addition, military clothing must be resistant to wear, tear, friction and are colored. The fibers obtained by you have extremely small mechanical properties (Figure S2). With such maximum fiber breaking forces (about 0.08N), how do you think the yarns or fabrics can be made to meet the conditions required by the protective equipment?
The use of silk as a textile support for JSLIST equipment is not financially justified either; probably the cost of such a military equipment would far outweigh the cost of the current ones that equips the armies from the world and which have cotton + polyester (50:50) as textile support.
Taking into account all my observations, I would suggest that you mention in the article that carbon-activated silk fibers could be used to make the lining of military equipment. The lining layer could be formed from a non-woven face laminated in activated carbon spheres and bonded to a knitted or woven back.
Author Response
We thank the reviewer and are grateful for their feedback. We have now included the following sentence in the conclusion (lines 179 – 182) to address their comments:
“Drawbacks of such fibres include relatively poor mechanical properties, difficulty of dying/colouration and higher costs compared to materials typically employed in military uniforms (e.g., 50:50 cotton and polyester blends); but these issues could potentially be mitigated by bonding/knitting the porous fabrics as backing or lining layers to existing established materials.”